# Longitudinal proteomic profiling of the inflammatory response in dengue patients

**Fadel Muhammad Garishah**[1,2], **Collins K. Boahen**[1,3], **Nadira Vadaq**[1,2], **Setyo G. Pramudo**[4,5], **Rahajeng N. Tunjungputri**[1,2], **Silvita Fitri Riswari**[1,6,7], **Ronald P. van Rij**[8], **Bachti Alisjahbana**[6,9], **Muhammad Hussein Gasem**[2,4], **André J. A. M. van der Ven**[1], **Quirijn de Mast**[1]*

1 Department of Internal Medicine and the Radboud Center for Infectious Diseases, Radboud University Medical Center, Nijmegen, The Netherlands, 2 Center for Tropical and Infectious Diseases (CENTRID), Faculty of Medicine, Diponegoro University, Dr. Kariadi Hospital, Semarang, Indonesia, 3 Department of Internal Medicine, Radboud Institute for Molecular Life Sciences, Radboud University Medical Center, Nijmegen, The Netherlands, 4 Department of Internal Medicine, Diponegoro National University Hospital, Faculty of Medicine, Diponegoro University, Semarang, Indonesia, 5 Department of Internal Medicine, William Booth Hospital, Semarang, Indonesia, 6 Research Center for Care and Control of Infectious Disease (RC3ID), Universitas Padjadjaran, Bandung, Indonesia, 7 Department of Biomedical Sciences, Faculty of Medicine, Universitas Padjadjaran, Bandung, Indonesia, 8 Department of Medical Microbiology, Radboud Institute for Molecular Life Sciences, Radboud University Medical Center, Nijmegen, The Netherlands, 9 Department of Internal Medicine, Hasan Sadikin General Hospital, Faculty of Medicine, Universitas Padjadjaran, Bandung, Indonesia

* Quirijn.deMast@radboudumc.nl

**Data Availability Statement:** All relevant data are within the manuscript and its Supporting Information files.

## Abstract

### Background

The immunopathogenesis of dengue virus (DENV) infection remains incompletely understood. To increase our understanding of inflammatory response in non-severe dengue, we assessed longitudinal changes in the inflammatory proteome in patients with an acute DENV infection.

### Methods

Using a multiplex proximity extension assay (PEA), we measured relative levels of 368 inflammatory markers in plasma samples from hospitalized patients with non-severe DENV infection in the acute (n = 43) and convalescence (n = 35) phase of the infection and samples of healthy controls (n = 10).

### Results

We identified 203 upregulated and 39 downregulated proteins in acute versus convalescent plasma samples. The upregulated proteins had a strong representation of interferon (IFN) and IFN-inducible effector proteins, cytokines (e.g. IL-10, IL-33) and cytokine receptors, chemokines, pro-apoptotic proteins (e.g. granzymes) and endothelial markers. A number of differentially expressed proteins (DEPs) have not been reported in previous studies. Functional network analysis highlighted a central role for IFNγ, IL-10, IL-33 and chemokines. We identified different novel associations between inflammatory proteins and circulating concentrations of the endothelial glycocalyx disruption surrogate marker syndecan-1.

**Funding:** FMG was financially supported by the Indonesian Endowment Fund for Education (LPDP) Scholarship from the Ministry of Finance Republic of Indonesia. Patient inclusion was financially supported by ZonMW (grant 451001005) to QM. The funders had no role in study design, data collection and analysis, decision to publish, or preparation of the manuscript.

**Competing interests:** The authors have declared that no competing interests exist.

**Conclusion:** This unbiased proteome analysis provides a comprehensive insight in the inflammatory response in DENV infection and its association with glycocalyx disruption.

## Author summary

Dengue virus (DENV) infection is the most common arboviral infection worldwide. Significant gaps remain in our understanding of the pathogenesis of dengue, including the changes that occur in the immune and inflammatory responses. In this present study, we used a multiplex proximity extension assay (PEA) from Olink Proteomics AB to assess levels of 368 inflammation-related proteins in plasma samples from the acute and convalescence phase of the infection in hospitalized adult patients with non-severe dengue. We show a pronounced increase in interferon (IFN) and IFN-related antiviral proteins, a selection of cytokines and cytokine receptors, members of the CCL and CXCL chemokine family, pro-apoptotic proteins and proteins involved in endothelial integrity in the acute phase of the illness. In addition, we found a large number of significant associations between these proteins and concentrations of the endothelial glycocalyx degradation marker syndecan-1. In summary, our proteome analysis provides a comprehensive insight in longitudinal changes in inflammation-related proteins in non-severe dengue and the association with endothelial glycocalyx perturbation.

## Background

Dengue is one of the most important arboviral infections, causing an estimated 100 million symptomatic infections annually worldwide [1]. The clinical manifestations of dengue infection range from a mild febrile illness to a life-threatening disease [2]. Dengue is classified into dengue with or without warning signs and severe dengue [3]. Patients with warning signs require strict observation and medical intervention. Only a small proportion of patients progress to severe disease, mostly characterized by a transient vascular hyperpermeability syndrome that may lead to fluid accumulation and shock, severe bleeding and organ impairment [3].

The immunopathogenesis of dengue virus (DENV) infection is only partially elucidated [4]. It is generally acknowledged that a dysregulated innate and adaptive immune response is a key factor in the complications of dengue [5]. Earlier studies have highlighted the upregulation of different cytokines, chemokines and lipid mediators in hospitalized dengue patients [6, 7]. However, our understanding of the immunological pathways in dengue remains incomplete as most studies measured only a limited number of inflammatory markers.

A comprehensive description of the changes in inflammatory pathways may help to better map the pattern of host defense responses during an acute DENV infection. In this study, we studied the expression of 368 inflammatory proteins using a Proximity Extension Analysis (PEA) technology (Olink Proteomics AB) in plasma of hospitalized patients with DENV infection in the acute and in the convalescent phase and in a small group of healthy volunteers. We also constructed a protein interaction network and explored associations of the differentially expressed proteins (DEPs) with participant characteristics, including signs of plasma leakage. Finally, disruption of the endothelial glycocalyx integrity is increasingly recognized as a key process in plasma leakage and, therefore, we determined associations of DEPs with plasma syndecan-1 concentrations, which is a surrogate marker of endothelial glycocalyx degradation.

## Methods

### Ethics statement

The Medical Research Ethics Committees of the Faculty of Medicine, Diponegoro University, Dr. Kariadi Hospital, Semarang, and Faculty of Medicine, Universitas Padjadjaran, Bandung, Indonesia approved the study (No:650/EC/FK-RSDK/XI/2017). The Medical Research Ethics Committee Arnhem-Nijmegen Region (NL32357.091.10) approved the collection of plasma from healthy volunteers. All subjects provided written informed consent prior to enrollment. All procedures were performed according to the Declaration of Helsinki.

### Study design and patients

This study used plasma samples from subjects in a phase 2, double-blind, multicenter, randomized placebo-controlled trial on the effect of oseltamivir phosphate on platelet recovery and plasma leakage in adults with dengue and thrombocytopenia (ISRCTN35227717) [8]. In brief, adult dengue patients ($\geq$16 years old) with fever ($\leq$ 6 days), thrombocytopenia ($< 70$ x$10^9$/L) and positive for rapid dengue virus (DENV) non-structural protein-1 (NS1) or IgM anti-DENV (PanBio, Diagnostics, Windsor, Australia) were enrolled in six different hospitals in Central and West Java, Indonesia, between January 2018 –July 2019. For the proteomics analysis, we included blood samples from the acute phase (n = 43) and paired convalescent samples (approximately 21 days post enrollment; n = 35). Patients with a positive NS1 or IgM anti-DENV and negative IgG anti-DENV during hospitalization were considered as probable primary dengue infection, the remainder were considered as probable secondary dengue infection. Thirty-one acute samples were from participants assigned to the placebo group of the trial and 12 from the oseltamivir group. Of the latter group, ten convalescence samples were included as well. All acute samples were taken prior to drug administration for both placebo and oseltamivir groups. Finally, ten plasma samples from adult healthy Indonesian volunteers were included, who were enrolled at Radboud university medical center, Nijmegen, The Netherlands.

### Sample processing and data collection

Plasma samples were obtained from 3.2% citrate-anticoagulated blood (BD Vacutainer, BD Biosciences, USA) centrifuged at 2061g for 15 minutes at room temperature to obtain platelet-poor plasma (PPP). All samples were stored at -80˚C until measurement.

### Proteomic analysis

Plasma proteins were measured with the Olink Explore 384 Inflammation panel using proximity extension technology (Olink Proteomics AB, Uppsala, Sweden) [9]. This panel includes 368 inflammation-related proteins. To enhance protein detectability and specificity, this assay utilizes the binding of target proteins by paired oligonucleotide antibody probes, followed by hybridization and amplification using a standard real-time PCR. Protein concentration are reported as normalized protein expression values (NPX), which is an arbitrary unit in a Log2 scale that is calculated from normalized Ct values. Validation data of the assay are available on the Olink website (www.olink.com). All samples were measured in the same batch in October 2020. Proteins were excluded from analysis when values were both below the detection limit in more than 25% of all samples and the difference in undetectable values between the acute and convalescence samples was less than 20%.

　　A functional network analysis was performed using the publicly available STRING database (version 11.5) [10] and visualized using Cytoscape version 3.9.0 [11]. We limited the analysis

to the top 50 DEPs between acute and convalescent samples. The minimum required interaction score was set to a medium confidence (0.4). Finally, to identify relevant pathways, we performed a pathway analysis of the 50 DEPs using the gene ontology database.

## Clinical and laboratory parameter measurements

The presence of ascites, pleural fluid and gall-bladder wall thickening was assessed daily using handheld ultrasonography (Phillips Lumify, Amsterdam, The Netherlands) with 6 hours fasting prior to the procedure. A complete blood count was performed twice daily using a standardized hematology analyzer in each study center. Circulating albumin and syndecan-1 concentrations were determined as previously described [8].

## Statistical analysis

Differential expression analysis was performed using a moderated paired t-test (acute versus convalescent samples (n = 35)) or a linear model (acute (n = 43) versus control (n = 10) samples) of NPX values using Limma R package [12], which employs an empirical Bayes method to moderate the standard errors of the fold change. Age, sex and BMI were statistically insignificant between patients and controls and were therefore not included in the linear model as covariates. Associations between variables were analyzed using Spearman's correlation analysis. The Benjamini-Hochberg multiple testing correction was performed in differential expression and correlational analyses with an FDR $p$-value $< 0.05$ considered to be statistically significant. All analyses and visualizations were performed using RStudio version 1.3 for Mac (RStudio, Boston, MA, USA). To further validate our findings, we intersected the DEPs identified in our study with differentially expressed transcripts (DETs) reported in earlier studies that compared gene profiles between acute versus convalescent dengue patients or controls or between patients with severe versus non-severe illness [13–24].

## Results

### Clinical Characteristics

Clinical characteristics of the patients and controls were summarized in **Table 1**. Plasma samples from 43 dengue patients were included in the analysis; for 35 patients, paired convalescent samples were available. In addition, plasma samples of 10 adult healthy Indonesian volunteers were used. The dengue patients were enrolled at a median of 5 days (IQR, 4–5) after symptom onset; 25 (58.1%) patients were classified as probable primary dengue infection and 18 (41.9%) as probable secondary infection. At enrollment, 26 (62%) patients had a thickened gallbladder wall (>3mm) and 15 (35.7%) patients had ascites and/or pleural fluid detected by ultrasonography. None of the participants fulfilled the World Health Organization (WHO) 2009 criteria for severe dengue and none died during follow-up [3]. In addition, using the WHO 2011 criteria [25], 25 (58%) of the dengue patients were classified as having dengue fever (DF) and 18 (42%) as dengue hemorrhagic fever (DHF) at study enrollment. At hospital discharge, 17 (40%) were classified as having had DF and 26 (60%) as DHF.

### Architecture of the plasma inflammatory proteome

Principal component analysis (PCA) of the inflammatory proteins (n = 337) revealed a clear separation between the acute dengue samples with convalescent dengue and control samples (all $p < 0.0001$) (**Fig 1A**). An unsupervised hierarchical clustering also suggested a similar finding (**S2 Fig**). Convalescent samples shifted towards the healthy controls. PCA analysis did not reveal a separation across samples of the different hospitals (**S1 Fig**). Expression analysis in

**Table 1. Clinical characteristics.**

| Variables | Dengue virus infection | | Controls |
| --- | --- | --- | --- |
| | Acute phase | Convalescent phase | |
| Number | 43 | 35 | 10 |
| Males | 30 (69.8) | 24 (68.6) | 4(40) |
| Age, years | 25 (19–36) | 25 (19–36) | 28 (27–34) |
| Days after symptom onset, days | 5 (4–5) | 26 (25–26) | - |
| BMI, kg/m$^2$ | 21 (19–24) | 22 (20–25) | 22 (21–25) |
| DENV-NS1 antigen positive, n (%) | 29/40 (72.5) | 25/33(75.8) | - |
| Anti-DENV IgM positive, n (%) | 20/43 (46.5) | 15/35 (42.9) | - |
| Anti-DENV IgG positive, n (%) | 18/43 (41.9) | 14/35 (40) | - |
| **Immune status, n (%)** | | | |
| Probable primary dengue infection | 25/43 (58.1) | 15/35 (42.9) | - |
| Probable secondary dengue infection | 18/43 (41.9) | 14/35 (40) | - |
| **Clinical laboratory tests** | | | |
| Hemoglobin, g/dL | 15.2 (13.5–16.4) | 12 (14.3–15) | 13.4 (12.3–15) |
| Hematocrit, % | 44.5 (39.6–48) | 41 (35–42) | 41 (36–46) |
| Leukocyte number, x10$^9$/L | 4.4 (3.5–6.1) | 6.4 (5.7–8) | 6.7 (5.3–7.4) |
| Platelet number, x 10$^9$/L | 39 (32–48) | 271 (235–329) | 205 (197–225) |
| Albumin, g/dL | 2.7 (2.5–3) | 3.4 (3.3–3.5) | - |
| Alanine transaminase, IU/L | 53 (35–86) | - | - |
| **Plasma leakage parameters at enrollment** | | | |
| Gallbladder wall thickening, n (%) | 26/42 (61.9) | - | - |
| Ascites and/or pleural fluid, n (%) | 15/42 (35.7) | - | - |
| **Plasma leakage parameters during hospitalization** | | | |
| Gallbladder wall thickening, n (%) | 35/42 (83.3) | - | - |
| Ascites or pleural fluid, n (%) | 23/42 (54.8) | - | - |
| Hemoconcentration, n (%) | 12/43 (27.9) | - | - |

Data are presented as median with interquartile range or number (n) with percentage (%). Ascites and pleural fluid accumulation were determined by a daily bedside ultrasonography. Gallbladder wall thickening was defined as a gallbladder wall thickness >3 mm. Hemoconcentration was defined as single hematocrit value of > 44% for female or >50% for male in acute phase of dengue. BMI = body mass index; DENV = dengue virus.

acute versus convalescent samples (n = 35) yielded 242/337 differentially expressed proteins (DEPs), of which 203 (84%) were significantly upregulated and 39 (16%) downregulated in the acute phase (**Fig 1C**). Expression analysis in acute samples (n = 43) compared to healthy control samples (n = 10) yielded 204 (93%) upregulated and 15 (7%) downregulated proteins (**S3A Fig**) with a strong overlap in DEPs in the acute vs convalescent sample analysis (**Figs 1B and S3B**). Overall, more inflammatory proteins were up- than downregulated and with a higher log fold-change. Ten convalescent samples were from participants allocated to the oseltamivir group, but there were no significant differences in protein concentrations in convalescent plasma from those allocated to the oseltamivir or placebo group (**S6 Fig**).

The top 50 up- and downregulated proteins in acute dengue samples were highlighted in a volcano plot (**Fig 1C**) and summarized in **Table 2**. A list of all significantly up- and downregulated proteins is given in **S1 Table**. The most prominently upregulated proteins were granzyme B (GZMB) and a number of interferon (IFN)-inducible effector proteins, including the intracellular antibody receptor tripartite motif-containing protein 21 (TRIM21), IFN-induced Protein 10 (IP-10; also known as chemokine C-X-C motif ligand 10 [CXCL10]), guanylate-binding protein 2 (GBP2), sterile alpha motif domain-containing 9-like (SAMDL9) and sialic

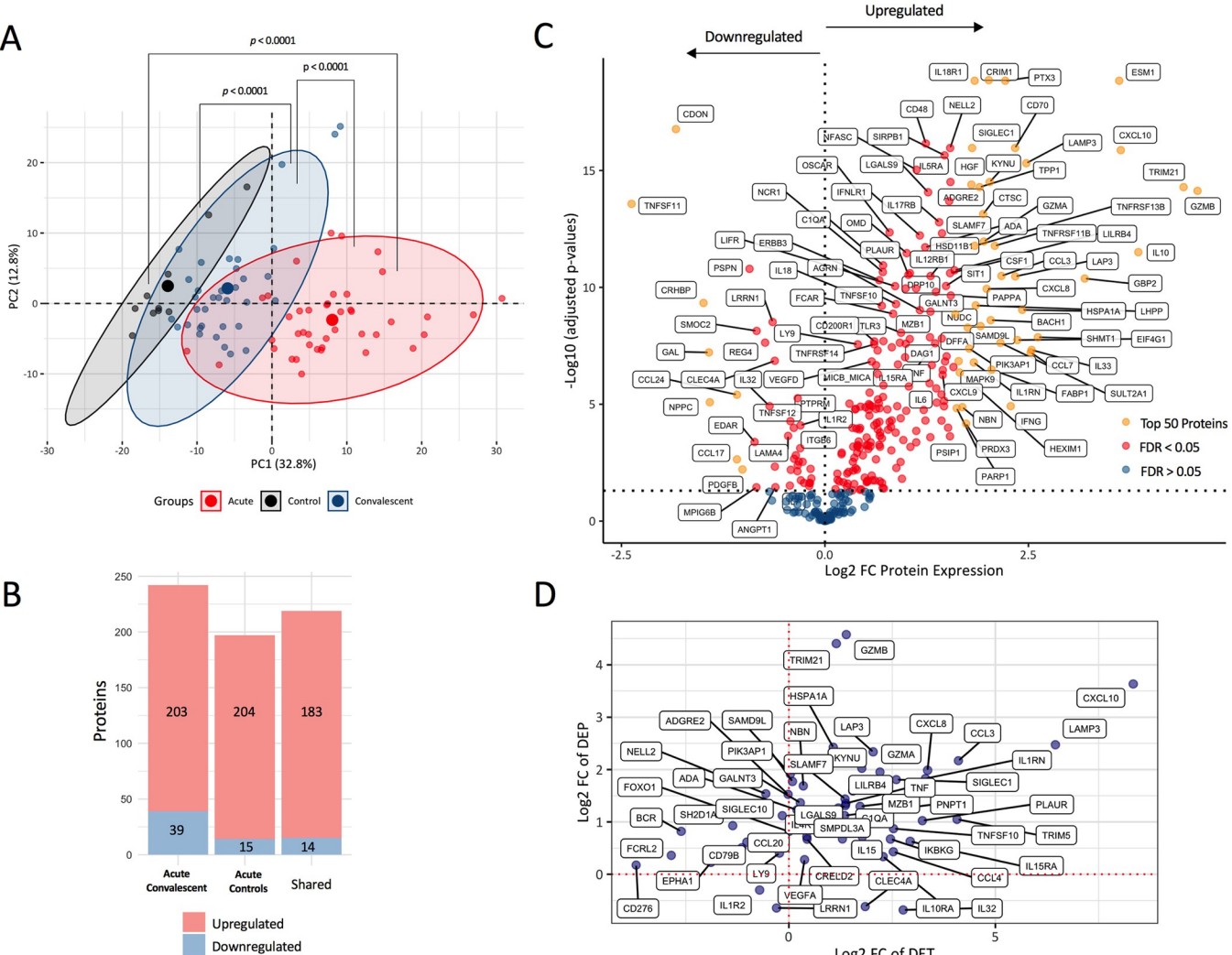

**Fig 1. Inflammatory protein profile and the differentially expressed proteins (DEPs) in dengue infection. (A)** Principal component analysis of acute dengue patients (N = 43), paired convalescent dengue patients (N = 35) and healthy controls (N = 10). The mean differences of PC1 between groups were analyzed using Mann-Whitney U test. Larger dots represent the central of cluster. **(B)** Bar chart depicting the numbers of up- and downregulated proteins in acute vs convalescent samples, acute vs control samples and their overlap. **(C)** Volcano plot displaying differential expression between paired acute versus convalescent samples of dengue patients (n = 35), showing the Log2 Fold-Change (Log2 FC) of normalized protein expression (NPX) on the X-axis and the (-Log10) of adjusted *p*-value (FDR) (Benjamini-Hochberg False-Discovery Rate) on the Y-axis. Gold = top 50 DEPs, Red = FDR < 0.05 and Blue = FDR > 0.05. **(D)**. Scatter plot displaying the Log2 FC values of the shared DEPs with previously identified differentially expressed transcripts (DETs) in acute versus convalescent dengue/healthy controls (N = 52).

acid binding immunoglobulin-like lectin 1 (SIGLEC1), also known as sialoadhesin and CD169 (**Figs 1C and 2**). The latter is a type I interferon-inducible receptor on monocytes that has been used as a surrogate marker for type I interferon activity [26]. The cytokines and cytokine receptors that were most markedly upregulated were interleukin (IL)-10, IL-33, IFN-gamma (IFNG), CXCL8 (also known as IL-8), IL-18 receptor-1 (IL18R1) and IL-1 receptor antagonist (IL1RN). Less markedly upregulated (Log2FC 0.55–1.37) cytokines and cytokine receptors were IL-6, tumor necrosis factor (TNF), IL-12 receptor beta 1 subunit (IL12RB1), IFN lambda receptor 1 (IFNLR1), IL-18, IL-17C and lymphotoxin alpha (LTA).

Other prominently upregulated proteins were lysosome-associated membrane glycoprotein 3 (LAMP-3) and costimulatory ligand CD70 [27, 28] and different growth factors, including

**Table 2. Top 50 differentially expressed proteins in acute dengue versus convalescent samples.**

| Protein | Log2FC | FDR | Name and selection of biological functions | Dengue Refs. |
|---|---|---|---|---|
| **Upregulated Proteins** | | | | |
| GZMB | 4.58 | 7.42E-15 | Granzyme B; Cytotoxic lymphocyte-derived serine protease which eradicates virus-infected cell through the induction of apoptosis. | [37] |
| TRIM21 | 4.41 | 5.16E-15 | Tripartite motif-containing protein 21 or E3 ubiquitin-protein ligase TRIM21; Intracellular antibody Fc domain effector which mediates virus proteolysis. | [38] |
| IL10 | 3.85 | 3.11E-12 | Interleukin-10; Anti-inflammatory cytokine. | [39] |
| CXCL10 | 3.63 | 1.35E-16 | C-X-C motif chemokine 10 or IFNγ-induced protein 10 (IP-10); Recruitment factor for monocytes, macrophages, T cells, NK cells and dendritic cells. | [40] |
| ESM1 | 3.62 | 1.44E-19 | Endothelial cell-specific molecule 1 (endocan); Endothelial cell-associated proteoglycan and modulator of angiogenesis and vascular permeability. | [41] |
| GBP2 | 3.19 | 4.18E-11 | IFN-induced guanylate-binding protein 2; IFN-inducible large GTPase with a role in host defense against intracellular microorganisms. | [18, 42] |
| SHMT1 | 2.62 | 1.36E-08 | Serine Hydroxymethyltransferase-1; Catalyzing serine conversion into glycine and folic acids and important in cell proliferation. | |
| IL33 | 2.54 | 4.74E-08 | Interleukin-33; Induction of pro-inflammatory state, NK cell activation and recruitment. | [43] |
| CCL7 | 2.53 | 6.08E-08 | C-C motif chemokine 7 or monocyte-chemotactic protein 3 (MCP3); Recruitment factor for monocytes, dendritic cells, neutrophils, NK cells and T cells. | [44] |
| LAMP3 | 2.47 | 4.88E-16 | Lysosome-associated membrane glycoprotein 3 or Dendritic cell lysosomal associated membrane glycoprotein; Lysosomal protein linked with dendritic cell maturation. | |
| HSPA1A | 2.42 | 9.17E-10 | Heat shock 70 kDa protein 1A; Cell chaperone, interferes with DENV replication. | [45] |
| EIF4G1 | 2.36 | 1.79E-08 | Eukaryotic translation initiation factor 4 gamma 1; Recognition and regulation of messenger RNA. | |
| LAP3 | 2.34 | 3.39E-11 | Cytosolic aminopeptidase; Regulator of cellular surface expression of MHC Class I. | |
| CD70 | 2.34 | 1.09E-16 | Cluster of Differentiation 70; CD27 costimulatory receptor which is highly expressed on activated lymphocytes. | [46] |
| IFNG | 2.28 | 1.21E-05 | Interferon gamma; Key regulator of cellular mediated immune responses. | [47] |
| PTX3 | 2.21 | 1.44E-19 | Pentraxin-related protein 3; Activates the complement system and opsonizes pathogens. | [48] |
| CCL3 | 2.17 | 3.24E-11 | C-C motif chemokine 3 or macrophage inflammatory protein 1-alpha; Recruitment factor for macrophages, monocytes and neutrophils. | |
| SULT2A1 | 2.16 | 2.36E-08 | Human cytosolic sulfotransferases 2A1; Catalyzes the sulfation of steroid precursors. | |
| TNFRSF11B | 2.09 | 1.67E-12 | Tumor necrosis factor receptor superfamily member 11B or osteoprotegerin; Modulation of osteoclastogenesis and T-cell activation. | [49] |
| FABP1 | 2.05 | 3.30E-07 | Fatty acid-binding protein-1; Role in lipid metabolism. | [50] |
| BACH1 | 2.04 | 2.52E-09 | BTB Domain and CNC Homolog 1; Transcription regulating protein. | [51] |
| KYNU | 2.02 | 3.10E-15 | Kynureninase; Key enzyme in tryptophan metabolic pathway. | |
| CRIM1 | 2.01 | 1.36E-19 | Cysteine-rich motor neuron 1 protein; Regulator of endothelial permeability through interaction with vascular endothelial growth factor (VEGF). | |
| CXCL8 | 1.99 | 1.16E-10 | C-X-C Motif Chemokine Ligand 8 or Interleukin-8; Chemotaxis of neutrophils. | [52] |
| GZMA | 1.95 | 1.09E-12 | Granzyme A; Cytotoxic lymphocyte-derived serine protease which eradicates virus-infected cell through the induction of apoptosis. | [53] |
| CTSC | 1.94 | 7.31E-14 | Cathepsin C; Activator of serine proteases in immune cells. | [54] |

(*Continued*)

**Table 2.** (Continued)

| Protein | Log2FC | FDR | Name and selection of biological functions | Dengue Refs. |
|---------|--------|-----|--------------------------------------------|--------------|
| SAMD9L | 1.91 | 4.55E-09 | Sterile alpha motif domain-containing protein 9-like; Regulator of cell proliferation and viral replication. | [18] |
| TPP1 | 1.90 | 5.16E-15 | Tripeptidyl-peptidase 1; Lysosomal enzyme. | |
| LHPP | 1.85 | 5.92E-10 | Phospholysine phosphohistidine inorganic pyrophosphate phosphatase; Histidine phosphatase. | |
| TNFRSF13B | 1.84 | 1.67E-12 | Tumor necrosis factor receptor superfamily member 13B or Transmembrane activator and Calcium-modulator and cyclophilin ligand interactor (TACI); Regulator of T and B cell interaction. | |
| IL18R1 | 1.84 | 1.44E-19 | Interleukin-18 receptor 1; Receptor for IL-18. | [55] |
| IL1RN | 1.83 | 1.63E-07 | Interleukin-1 receptor antagonist; Anti-inflammatory cytokine through a competition binding to IL-1 receptor (IL1R). | [40] |
| SIGLEC1 | 1.80 | 1.09E-16 | Sialic Acid Binding Ig Like Lectin 1 or Sialoadhesin or CD169; Type I interferon-inducible pattern recognition receptor on monocytes and macrophages. | |
| HGF | 1.79 | 4.01E-15 | Hepatocyte Growth Factor; Mitogen for hepatocytes and other cells, including endothelial cells | [40] |
| PIK3AP1 | 1.77 | 4.11E-08 | Phosphoinositide 3-kinase adapter protein 1; B-cell receptor (BCR) phosphoinositide 3-kinase (PI3K)-Akt signaling pathway and BCR co-receptor of CD19. | [18] |
| DFFA | 1.75 | 5.27E-09 | DNA fragmentation factor subunit alpha; Substrate for caspase-3 that triggers DNA fragmentation during apoptosis. | |
| PARP1 | 1.73 | 6.78E-05 | Poly (ADP-ribose) polymerase 1; Nuclear enzyme that plays a role in different cellular biological processes, including DNA repair, and gene transcription of inflammatory mediators | |
| NBN | 1.69 | 1.33E-05 | Nibrin; Cellular response to DNA damage. | |
| HEXIM1 | 1.66 | 4.39E-07 | Hexamethylene Bisacetamide Inducible 1; Inhibitor of RNA polymerase II transcription elongation. | |
| EGLN1 | 1.64 | 1.43E-07 | Hypoxia-inducible factor prolyl hydroxylase 2; Regulates the stability of Hypoxia-Inducible Factor 1. | |
| PRDX3 | 1.62 | 1.47E-05 | Peroxiredoxin 3; Peroxidase acting as a cellular antioxidant. | |
| NUDC | 1.61 | 1.41E-09 | Nuclear migration protein; Mitotic factor of hematopoietic cells. | |
| **Downregulated Proteins** | | | | |
| TNFSF11 | -2.38 | 2.67E-14 | Tumor necrosis factor ligand superfamily member 11, also known as Receptor activator of nuclear factor kappa-B ligand (RANKL); Member of the TNF cytokine family. Osteoclast differentiation, Regulator of interaction between dendritic cells and T-cells and T-cell-dependent immune responses. | |
| CDON | -1.83 | 1.70E-17 | Cell adhesion molecule-related/down-regulated by oncogenes; Co-receptor ligand of Patched-1 (PTCH1) which decreases endothelial barrier function through hedgehog signaling pathway. | |
| CRHBP | -1.49 | 4.65E-10 | Corticotropin-releasing factor-binding protein; Binds and inactivates corticotropin-releasing factor (CRF). | |
| GAL | -1.43 | 6.08E-08 | Galanin peptides; Neuropeptide with immunomodulatory properties. | |
| NPPC | -1.42 | 8.24E-06 | C-type natriuretic peptide precursor; Endothelial-derived molecule which regulates vasodilatation, blood pressure and inhibits leukocyte recruitment. | |
| CCL17 | -1.08 | 2.29E-03 | C-C motif chemokine 17; Recruitment factor for T regulatory cells. | [56] |
| CCL24 | -1.08 | 3.93E-06 | C-C motif chemokine 24; Recruitment factor for eosinophils. | |
| PDGFB | -1.01 | 6.18E-03 | Platelet-derived growth factor subunit B | [44] |

*Relevant and specific references for the protein as demonstrated in dengue patients or *ex vivo* and *in vitro* experiments with dengue virus (DENV) or DENV-related antigen stimulation.

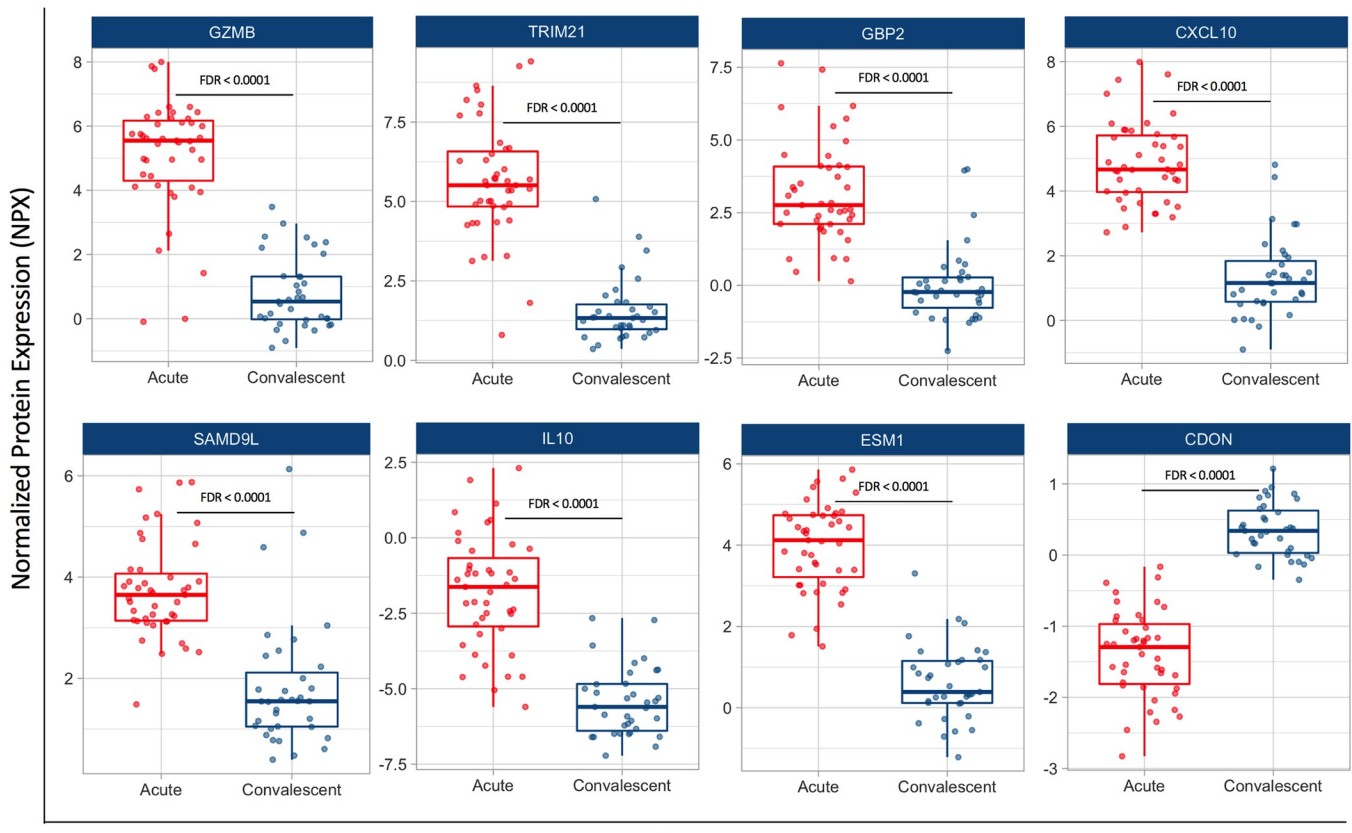

**Fig 2. Selection of the top differentially expressed proteins (DEPs) in the acute phase compared to convalescence.** Depicted are individual data together with a box plot showing median with interquartile range and adjusted *p*-value (FDR) (Benjamini-Hochberg False-Discovery Rate). GZMB, Granzyme B; TRIM21, Tripartite motif-containing protein 21; GBP2, Guanylate-Binding Protein 2; SAMD9L, Sterile Alpha Motif Domain-containing protein 9-Like; IL10, Interleukin-10; ESM1, Endothelial Cell-Specific Molecule 1; CDON, Cell adhesion molecule-related/Down-regulated by Oncogenes.

hepatocyte growth factor (HGF) and colony stimulating factor 1 (CSF1). Endothelial dysfunction is a key factor in dengue pathology and different endothelial markers were upregulated, including endothelial cell-specific molecule 1 (ESM1; also known as endocan), cysteine rich transmembrane BMP regulator 1 (CRIM1), which is involved in endothelial maintenance and integrity [29], and vascular endothelial growth factor D (VEGFD). Finally, different proteases that are known to play a key role in the antiviral immune response were upregulated. These included granzyme B and A (GZMB and GZMA) and cathepsin C (CTSC) which acts as a central coordinator for the activation of serine proteases in immune cells [30], lysosomal tripeptidyl-peptidase 1 (TPP1), which is important for lysosomal molecule degradation [31] and cytosolic aminopeptidase (LAP3), which is important for antigen processing and presentation [32].

Among the most downregulated proteins were Tumor necrosis factor ligand superfamily member 11 (TNFSF11; also known as Receptor activator of nuclear factor kappa-B ligand [RANKL]) and cell adhesion molecule-related/down-regulated by oncogenes (CDON). TNFSF11 belongs to the TNF cytokine family and is a regulator of osteoclastogenesis and of different immune functions, including the interaction between dendritic cells and T-cells and T-cell-dependent immune responses [33]. CDON was recently identified as a negative regulator of hedgehog signaling-driven endothelial integrity [34]. Other downregulated proteins

were corticotropin-releasing hormone binding protein (CRHB), which modulates corticotropin releasing factor (CRH), the neuropeptide and natural killer cell modulator galanin (GAL) [35] and endothelial C-type natriuretic peptide precursor (NPPC), which is important in fluid homeostasis [36].

## Validation of the plasma inflammatory proteome

Intersection of the 242 DEPs identified in our study with 1248 differentially expressed transcripts (DETs) reported in earlier studies comparing gene profiles between acute and convalescent dengue samples [13–24], showed that 53 (21.9%) DEPs/DETs were shared (**S7A Fig and S2 Table**), of which 38 (73%) in a similar direction (**Fig 1D**). C-X-C motif chemokine ligand 10 (CXCL10), a chemotactic factor for NK and T cells, was identified as a DET in most studies. Most of the shared DEPs/DETs correlated with syndecan-1 (SDC1) concentrations or clinical parameters (**S2 Table**). According to the gene ontology database, the shared DEPs/DETs were mainly involved in lymphocyte chemotaxis (CXCL10, CCL3, CCL4), interleukin-15 mediated signaling (IL15, IL15RA) and negative regulation of host viral transcription (TRIM21, LGALS9, CCL3, CCL4) (**S3 Table**). In addition, intersecting the DEPs with DETs associated with severe versus non-severe illness (n = 313) yielded 15 (6.2%) shared DEPs/DETs (**S7B Fig** and **S2 Table**).

## Functional network analysis of the inflammatory proteome

To further increase our insight in the functional interactions between the top DEPs, we constructed a protein-protein association network, using the STRING database [10]. We limited the analysis to the top 50 DEPs with a cutoff Log2 fold-change of > 1.5 for upregulated proteins (n = 42) and >1 for downregulated proteins (n = 8). This analysis yielded an enriched network with IFNγ as a central protein, along with other cytokines (IL10 and IL33) and members of the CXC chemokine ligand (CXCL8/10) and CC chemokine ligand (CCL3/7) family (**Fig 3**). Next, using the gene ontology biological processes term, we highlighted several important pathways involved in our protein-protein interaction network including response to IFNγ, regulation of viral life cycle, regulation of cell death and chemokine-mediated signaling pathway.

## Associations of the inflammatory proteins with clinical and laboratory profiles

Next, to understand the possible relationship between the inflammatory proteins and clinical outcomes, we assessed the associations of the DEPs with markers of dengue severity. Overall, there were no differences in levels of inflammation-related proteins between patients classified as DF and those with DHF (**S4A and S4B Fig**). Age, sex and BMI are associated with a higher risk for dengue complications [2, 57, 58]. In our cohort, especially age was associated with different inflammatory markers including TNF, IL1RN, chemokines and urokinase receptor (PLAUR) (**Fig 4**). Participants classified as having a probable secondary dengue infection had higher concentrations of different molecules involved in the host immune response, including lymphocyte function (CEACAM21, FOXO1, MZB1, PARP1, PPP1R9B, PSIP1, SH2D1A), glucocorticoid hormone regulation (CRHBP) and megakaryocyte maturation (MPIG6B). In contrast, no differences were observed in concentrations of pro-inflammatory cytokines, interferon-inducible effector proteins or granzymes. Overall, there were only few significant associations between DEPs and the following markers of dengue severity: platelet number, hematocrit, plasma albumin, gallbladder wall thickness and the presence of ascites or pleural fluid. Participants with ascites and/or pleural effusion during hospitalization had a trend for a

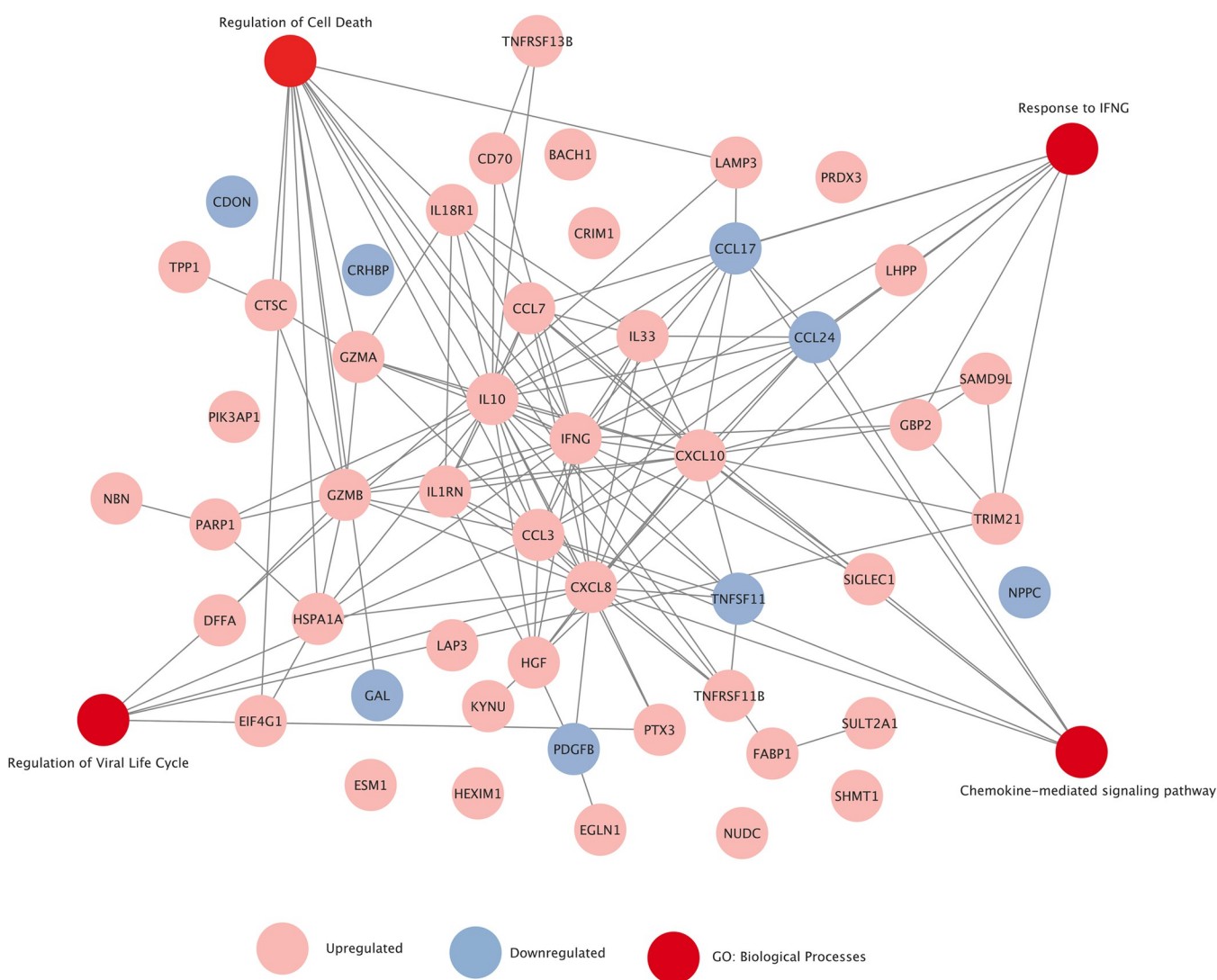

**Fig 3. Functional network analysis of the top 50 differentially expressed proteins in dengue patients.** Depicted is the protein-protein Interaction (PPI) network from the STRING database. The analysis was limited to the top 50 DEPs. The pink (upregulated) and blue (downregulated) nodes denote the proteins while the lines denote the relationships. PPI enrichment *p*-value was < 1x10^-16, indicating that the network has significantly more interactions than expected and that the proteins likely engage in biologically relevant interactions. Additional nodes of relevant pathways (red nodes) derived from gene ontology biological process term were included in the network.

higher ESM1 value (**S5A Fig**). In contrast, plasma concentrations of syndecan-1 (SDC1), a surrogate marker of endothelial glycocalyx degradation [59], correlated positively with a large number of inflammatory proteins (**Fig 4**) and fluid accumulation during hospitalization (**S5B Fig**). The strongest associations were observed for HGF ($r_s$ = 0.74; FDR 1.18x10^-6) and IL18R1 ($r_s$ = 0.71; FDR 6.59x10^-6). The heparanase/SDC1 axis is known to regulate HGF expression and function [60, 61]. SDC1 concentrations were also positively correlated with a number of other endothelium-related proteins, including glycocalyx-associated proteoglycans (agrin [AGRN] and ESM1), CRIM1, and osteoprotegerin/TNFRSF11B). Leukocyte number was negatively associated with a number of inflammatory proteins, including cytokines (IL10, IL15, IFNG, LTA, IL33, CSF1), IFN-stimulated genes (TRIM21), immune receptors (LIFR, IL18R1, IL22RA1, SIGLEC1) and chemokines (CCL4, CXCL8). Finally, plasma concentrations of the

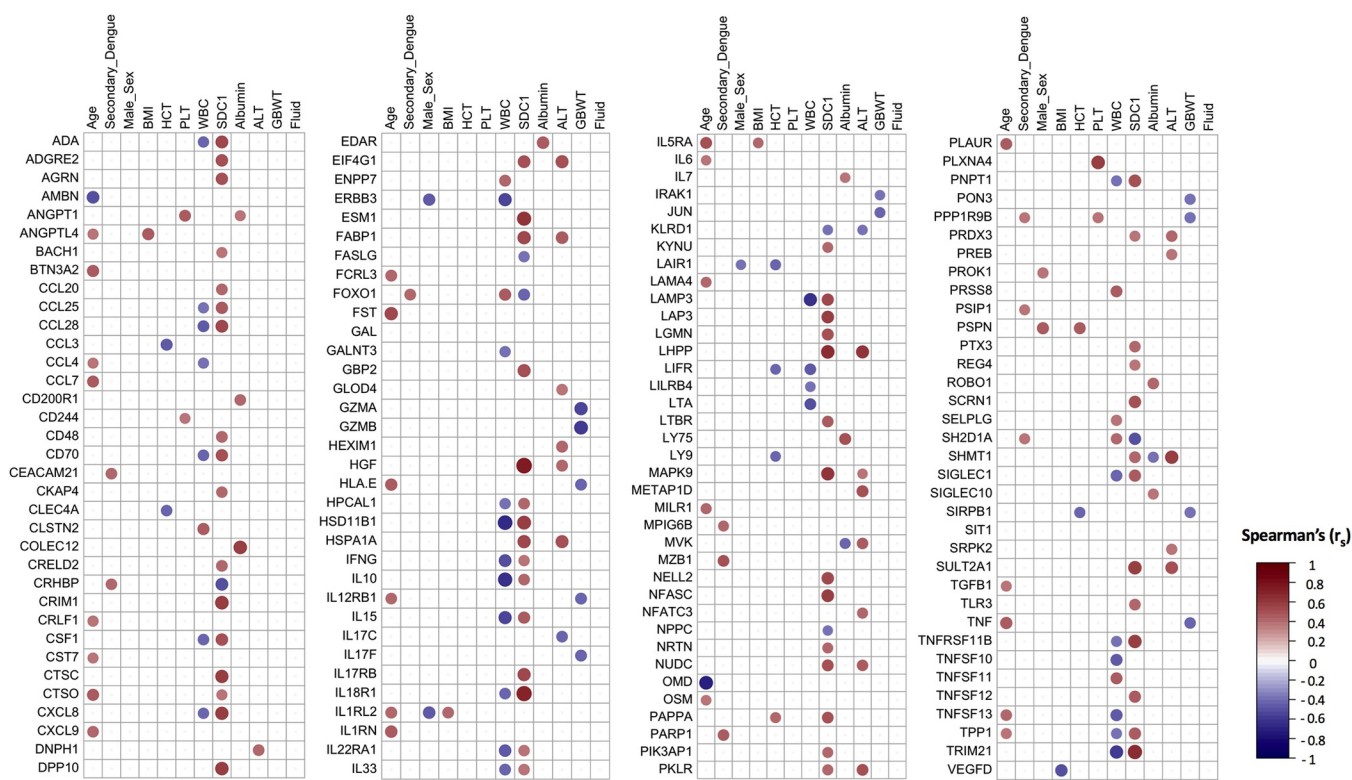

**Fig 4. Associations of the differentially expressed proteins with clinical and laboratory profiles.** Heatmap of associations between DEPs (N = 242) (normalized protein expression (NPX)) of 43 dengue patients with clinical and laboratory parameters at enrollment. Shown are the significant Spearman's correlation coefficients ($r_s$) after Benjamini-Hochberg false-discovery rate (FDR) correction $p$-value < 0.05. Bullet size and darker colors represent the stronger associations whereas red and blue corresponds to positive and negative associations. Age (years), Male Sex (Male), HCT, hematocrit; WBC, white blood cell number; PLT, platelet number; ALT, alanine transferase; SDC1, plasma syndecan-1; Secondary dengue; probable secondary dengue infection; GBWT, gall-bladder wall thickness; Fluid, presence of ascites and/or pleural fluid.

liver enzyme alanine transferase were positively associated with several proteins related to lipid and cholesterol metabolism, such as FABP1, MVK and SULT2A1.

## Discussion

Our unbiased proteomics analysis in plasma samples from DENV infection patients demonstrates activation of important components of the antiviral immune responses in acute DENV infection and a resolution thereof in the convalescent phase. A total of 203 of 337 proteins were upregulated with a strong representation of IFN and IFN-related antiviral proteins (e.g. GBP2, TRIM21 and SAMD9L), cytokines and cytokine receptors (e.g. IL10, IL33, IL1RN and IL18R1), members of the CCL and CXCL chemokine family (e.g. CXCL10, CCL7, CCL3, CXCL8, CCL25 and CXCL9), pro-apoptotic proteins (e.g. GZMA, GZMB, CTSC, TPP1) and proteins involved in endothelial integrity (e.g. ESM1, CRIM1 and VEGFD). Disruption of the endothelial glycocalyx is increasingly recognized to play a central role in the vascular complications of DENV infection [62]. The pathways that are involved in this disruption are incompletely understood. Our study and validation analysis revealed a large number of associations between the glycocalyx disruption marker SDC1 with inflammatory and endothelial proteins. Our results therefore suggest that the inflammatory environment contributes to glycocalyx dysfunction in DENV infection.

IFN-mediated innate immune responses are a key process in the first line host defense against viral infections. Our study shows the upregulation of different IFN-stimulated genes

(ISGs) that are likely important in orchestrating the host response against DENV replication. Guanylate-binding protein-2 (GBP2) is a member of the IFN-inducible guanosine triphosphatase (GTPase) family. GBP2 was shown to exert broad antiviral activity, including against Zika virus [63]. GBP2 was also part of a 20-gene set that was strongly associated with the progression to severe dengue [42]. In addition, the cytosolic ubiquitin ligase and antibody receptor TRIM21 was among the top upregulated DEPs and validated proteins. TRIM21 rapidly neutralizes antibody-bound viruses [64, 65], suggesting that it may play a role in neutralization and possibly antibody-dependent enhancement of DENV. In addition, TRIM21 is a negative regulator of proinflammatory cytokines and type I IFN production [66]. Our findings are consistent with a recent study that showed that the expression of TRIM21 was upregulated in helper innate lymphoid cells of patients with dengue hemorrhagic fever [38].

Our findings further reinforce the central roles for IL-10, CXCL10 and IL-33 in the host response to DENV infection. IL-10 and CXCL10 were previously shown to be markers of dengue severity and plasma leakage [39, 40, 67]. IL-33 is a member of the IL-1 family and, unlike most other cytokines, is normally located in the nucleus. It is released by various cells, including fibroblasts and epithelial, endothelial and immune cells as an alarmin following cell injury. IL-33 has been implicated in allergy, asthma and parasitic diseases, but also increasingly in chronic inflammatory conditions such as arthritis [68]. In a mouse model of DENV infection, IL-33 production was increased and shown to exert a disease-exacerbating role [43]. These findings, as well as the findings reported here, are in seeming contrast to previous studies that reported similar or even decreased plasma concentrations of IL-33 concentrations in patients with DENV infection compared to controls [69, 70], which is most-likely due to sensitivity of the currently available assays [71].

Another important finding is the upregulation of different proteases, including LAP3 and cathepsin C (CTSC). The latter is a lysosomal cysteine protease that activates cytotoxic lymphocyte-derived serine proteases, including granzyme A and B, and as such plays a role in the clearance of virus-infected cells [72]. In fact, granzyme B was the most strongly upregulated protein in our study. Granzyme B is stored in granules of cytotoxic T-cells and Natural Killer cells, both known to kill virally infected cells [73]. Cathepsins are also among the potent activators of heparan-degrading enzymes (e.g. heparanase and sialidase) and therefore may contribute to degradation of endothelial glycocalyx in dengue [54].

A transient vascular hyperpermeability syndrome is a key feature in dengue pathophysiology. The targeted proteomics panel also included different endothelial proteins. One of the most upregulated proteins in our study, ESM1 or endocan, was recently shown to be induced by DENV non-structural protein-1 (NS1) and suggested to be a promising biomarker for endothelial activation [41]. In addition, the upregulation of agrin proteoglycan and CRIM1 [74, 75] and the downregulation of CDON are counterregulatory mechanisms that limit plasma leakage [34]. TNFSF11/RANKL was the most pronounced downregulated protein, whereas its physiological inhibitor osteoprotegerin (TNFRSF11B/OPG) was upregulated, as previously shown by our group [49]. Besides bone homeostasis, the RANK-RANKL-OPG axis is increasingly recognized to play a key role in both the adaptive and innate immune system [76]. A number of inflammation-related proteins were associated with the endothelial glycocalyx degradation marker SDC1. However, the lack of associations between inflammatory proteins and clinical signs of dengue disease severity (e.g. HCT, fluid accumulation) may suggest that the differentially expressed proteins reflect their role in infection control rather than severity.

A specific strength of our study is the high number of inflammatory proteins in the targeted proteomics platform, which allows a comprehensive analysis of the host immune response in DENV infection and discovery of new biomarkers. The proximity extension assay, which

combines both antibody-binding and next-generation sequencing, is considered to be more sensitive and specific in comparison to mass-spectrometry proteomics analysis or other immunoassays such as ELISA [77]. In addition, we analyzed longitudinal samples of the participants, which limits interindividual variation, and participants were well characterized, including daily evaluation of fluid accumulation by ultrasonography. Limitations of our study are the fact that the number of participants was limited, and the participants were relatively homogenous in terms of severity of illness. This precludes the evaluation of the performance of the proteins identified here in the prediction of severe illness. Second, high viral loads have been reported to be associated with disease severity in some studies, but viral loads were unavailable in our study. Third, ten convalescence samples were from participants allocated to the oseltamivir group. Even though we cannot rule out potential secondary effects of the drug, there were at least two weeks between the last dose of oseltamivir and collection of convalescent plasma. In addition, DEP analysis did not reveal significant differences between samples from participants allocated to the oseltamivir or placebo group. Finally, despite the large number of target proteins in the panel, it inevitably lacks proteins that may be relevant in DENV infection, including type I interferons.

In summary, this proteomics study provides a comprehensive insight and linkage of the host inflammatory response and endothelial glycocalyx degradation in hospitalized patients with DENV infection. A number of up- and downregulated proteins have not been reported in previous studies, which shows the potential for the unbiased discovery of candidate biomarkers and predictors of severe disease using these novel technologies.

## Supporting information

**S1 Table. List of the differentially expressed proteins (DEPs).**
(DOCX)

**S2 Table. Shared differentially expressed proteins (DEPs) and previously identified transcripts (DETs).**
(DOCX)

**S3 Table. Top 15 Gene Ontology Biological Process functional enrichments of the 53 shared DEPs/DETs.**
(DOCX)

**S1 Fig. Principal component analysis of individual acute dengue patients across different hospitals.**
(DOCX)

**S2 Fig. Hierarchical clustering analysis.**
(DOCX)

**S3 Fig. Overlapped differentially expressed proteins (DEPs) between acute vs convalescent dengue and acute dengue vs healthy controls.**
(DOCX)

**S4 Fig. Differentially expressed proteins (DEPs) between dengue hemorrhagic fever (DHF) vs dengue fever (DF) patients.**
(DOCX)

**S5 Fig. Circulating Endothelial Cell Specific Molecule 1 (ESM1) and Syndecan-1 (SDC1) on acute phase in the presence of fluid accumulation during hospitalization.**
(DOCX)

**S6 Fig. Differentially expressed proteins (DEPs) in convalescent phase between Oseltamivir and Placebo arm.**
(DOCX)

**S7 Fig. Shared differentially expressed proteins (DEPs) and previously identified transcripts (DETs) from dengue patients.**
(DOCX)

## Acknowledgments

We acknowledge the contribution of all patients and healthy volunteers in this study. From the research site in Semarang, we acknowledge Prof. dr. Sultana M. H. Faradz (Center for Biomedical Research (CEBIOR), Faculty of Medicine, Diponegoro University) for her generous support of our work and the laboratory facilities, Evi Nurwulan for the laboratory assistance and also the support of Integrated Laboratory Unit (UPT) of Diponegoro University. From the research site in Bandung, we acknowledge dr. Haneng Marissangan, SpPD, dr. Any Yuliani, SpPK, M.Kes and their staffs (Dr. M. Salamun Air Force Hospital in Bandung) to support this study. We also acknowledge the support from the Immunology Laboratory, Faculty of Medicine, Universitas Padjadjaran, and Siti Rasnawati Mony, Dera Darmayanti, Tri Kusniati, and Qireneu Dwi Putri for providing technical assistances. Lastly, we also thank the contributions of all the research assistants in this study.

## Author Contributions

**Conceptualization:** Fadel Muhammad Garishah, Muhammad Hussein Gasem, André J. A. M. van der Ven, Quirijn de Mast.

**Data curation:** Fadel Muhammad Garishah, Collins K. Boahen, Quirijn de Mast.

**Formal analysis:** Fadel Muhammad Garishah, Collins K. Boahen, Nadira Vadaq, Quirijn de Mast.

**Funding acquisition:** Fadel Muhammad Garishah, Quirijn de Mast.

**Investigation:** Fadel Muhammad Garishah, Collins K. Boahen, Setyo G. Pramudo, Rahajeng N. Tunjungputri, Silvita Fitri Riswari, Quirijn de Mast.

**Methodology:** Fadel Muhammad Garishah, Collins K. Boahen, Nadira Vadaq, Rahajeng N. Tunjungputri, Quirijn de Mast.

**Project administration:** Fadel Muhammad Garishah, Setyo G. Pramudo, Rahajeng N. Tunjungputri, Silvita Fitri Riswari, Quirijn de Mast.

**Resources:** Fadel Muhammad Garishah, Collins K. Boahen, Setyo G. Pramudo, Rahajeng N. Tunjungputri, Silvita Fitri Riswari, Bachti Alisjahbana, Muhammad Hussein Gasem, Quirijn de Mast.

**Software:** Fadel Muhammad Garishah, Collins K. Boahen.

**Supervision:** Bachti Alisjahbana, Muhammad Hussein Gasem, André J. A. M. van der Ven, Quirijn de Mast.

**Validation:** Fadel Muhammad Garishah, Collins K. Boahen, Quirijn de Mast.

**Visualization:** Fadel Muhammad Garishah, Collins K. Boahen, Nadira Vadaq, Quirijn de Mast.

**Writing – original draft:** Fadel Muhammad Garishah, Quirijn de Mast.

**Writing – review & editing:** Fadel Muhammad Garishah, Collins K. Boahen, Nadira Vadaq, Setyo G. Pramudo, Rahajeng N. Tunjungputri, Silvita Fitri Riswari, Ronald P. van Rij, Bachti Alisjahbana, Muhammad Hussein Gasem, André J. A. M. van der Ven, Quirijn de Mast.

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
