## [Decision Letter · Decision Letter 0]

15 Sep 2022

Dear Mr. Garishah,

Thank you very much for submitting your manuscript "Longitudinal proteomic profiling of the inflammatory response in dengue patients" for consideration at PLOS Neglected Tropical Diseases. As with all papers reviewed by the journal, your manuscript was reviewed by members of the editorial board and by several independent reviewers. In light of the reviews (below this email), we would like to invite the resubmission of a significantly-revised version that takes into account the reviewers' comments. 

We cannot make any decision about publication until we have seen the revised manuscript and your response to the reviewers' comments. Your revised manuscript is also likely to be sent to reviewers for further evaluation.

Sincerely,

Simon Rayner

Academic Editor

Sunit Singh

Section Editor

Reviewer's Responses to Questions

**Key Review Criteria Required for Acceptance?**

**Methods**

-Are the objectives of the study clearly articulated with a clear testable hypothesis stated?

-Is the study design appropriate to address the stated objectives?

-Is the population clearly described and appropriate for the hypothesis being tested?

-Is the sample size sufficient to ensure adequate power to address the hypothesis being tested?

-Were correct statistical analysis used to support conclusions?

-Are there concerns about ethical or regulatory requirements being met?

Reviewer #1: I am not a statistician, however, based on what I am seeing, the sample size seems adequate to me and the statistical analysis supports the data. No ethical concerns.

Reviewer #2: Major points:

The study holds some potential since the patient cohorts is well defined, longitudinal and multicenter. The data appear adequately analyzed (including multiple hypothesis testing and some modelling; see i.e. LIMMA); however in its current stage the study appears rather preliminary and even the bioinformatic analysis is rather superficial in scope for a descriptive study, thereby limiting its relevance for both clinicians or virologists.

The authors present the outcome of a commercially-provided service (Olink technology), identifying a number of expected and unexpected differentially regulated proteins in the plasma (i.e. PPP) of both covalescent and severe patients. 

While this approach has certain advantages highlighted by the authors, it does not represent a novelty or an incremental benefit over other plasma proteomics methods, which have currently reached 3X the depth of the method used (see ie. https://doi.org/10.15252/emmm.201910427). Nonetheless, the identified proteins reported herein might be of great interest or relevance if they would be validated in dowstream experiments either in vitro or in silico. 

For some of the proteins robustly up- or down-regulated and not directly involved in cytokine responses (i.e. GBP2, HEXIM1, LAP3, etc..), it would be interesting and important to assess their relevance in vitro (i.e. upon overexpression/silencing in monocytes or macrophages) or confirm their up- or down-regulations by alternative approaches in clinical samples. Similarly, it would be interesting to follow up on 1 or 2 candidate proteins exhibiting high correlation with certain clinical parameters (i.e. HGF or IL18R1, Fig 4). Alternatively, if the authors aim to deliver a resource paper for the community, it would be important to provide thorough intersection of these results with all or most of the other published studies on plasma proteins profiling of DENV patients .

**Results**

-Does the analysis presented match the analysis plan?

-Are the results clearly and completely presented?

-Are the figures (Tables, Images) of sufficient quality for clarity?

Reviewer #1: Results are clearly presented and discussed

Reviewer #2: (No Response)

**Conclusions**

-Are the conclusions supported by the data presented?

-Are the limitations of analysis clearly described?

-Do the authors discuss how these data can be helpful to advance our understanding of the topic under study?

-Is public health relevance addressed?

Reviewer #1: Conclusions are supported by the data. A few minor limitations were not discussed (please see comments below).

Reviewer #2: (No Response)

**Editorial and Data Presentation Modifications?**

Reviewer #1: Minor revision

Reviewer #2: Minor points:

- As it currently stands, it is hard to infer the significance of the findings considering the rather small fold-change of most of the identified proteins. For instance in Fig.1, FDR<0.05, red labels: most of these proteins are differentially within 2 and 3-fold change to controls, yet highlighted as up-regulated. Which cut-off criteria was applied to fold-change for hit-calling?

-lines 365-367: “it inevitable lacks” please correct

-Mat. And methods, lines 119-134: Olink data processing and analysis: please provide more details. As the assay is not a commonly-used one, the concept behind the DAN barcoding and the data normalization and quantification should be provided.

- Figure 1A: How was the statistical significance of the different groups within the PCA calculated? Is this adjusted or unadjusted p-values presented?

- Supplementary Figure S4: please rescale the y-axis to display dynamic window of the data more accurately (i.e. p-values compressed towards x-axis).

**Summary and General Comments**

Reviewer #1: The manuscript by Garishah et al. aims at improving the understanding of the changes in inflammatory pathways and the pattern of host defense responses during acute dengue virus (DENV) infection. Based on a Proximity Extension Analysis (PEA) applied to longitudinal patient-derived plasma samples, the authors report an unbiased proteomic characterization wherein over 200 inflammatory proteins were found to be upregulated in the acute phase of dengue infection relative to the convalescent phase.

The novelty of the study lies in the associations observed between interferon-mediated pathways and other inflammatory proteins to circulating concentrations of syndecan-1, a surrogate marker for endothelial glycocalyx disruption. While it is hypothesized that vascular complications during DENV infection are directly affected by disruption of the endothelial glycocalyx, the inflammatory pathways involved in this process are to date not fully understood. The use of the PEA technology is also quite novel, and it confers an enhanced specificity and sensitivity than other techniques like mass spectrometry, with the advantage that a high number of inflammatory proteins can be analyzed simultaneously. 

While a major limitation of the study is that it does not include patients who progressed to severe dengue, it does contribute to the understanding of the host immune responses to uncomplicated dengue infection. Moreover, it offers an opportunity for the discovery and validation of new disease biomarkers. 

The manuscript is generally well-written. Some comments that the authors should address are listed below:

Comment 1: It is well established that antibody-dependent enhancement and secondary infections play a major role in the pathogenesis of dengue infection. Ideally, it would be important to correlate the expression level of the identified differentially expressed proteins with prior dengue exposure (primary vs. secondary infection).

Comment 2: High viral loads have been reported in some studies to be associated with disease severity. It would be interesting to assess and correlate viral loads and the differentially expressed proteins, together with the clinical parameters described in the study. 

Comment 3: This study was performed with a cohort of patients participating in a randomized trial evaluating the effect of oseltamivir on platelet recovery and plasma leakage in adults with dengue and thrombocytopenia. While acute samples were taken prior to drug administration, convalescent samples were taken afterwards. Potential secondary effects of the drug should therefore be considered and discussed as a limitation of the study.

Comment 4: The genetic background of the control subjects included in this study (from the Netherlands) was different from that of the patients (Indonesia). This should be mentioned as a limitation in the discussion.

Reviewer #2: This study by Garishah and colleagues investigates differentially-regulated proteins within the plasma of covalescent and severe Dengue patients in a multicenter longitudinal cohort, using a proprietary proteomics technology (Olink). The authors identify a number of up- and down-regulated proteins, presenting a STRING network analysis and correlation analysis of some of these proteins with known clinical profiles.

The study is well-written, the clinical data collection and the composition of the cohorts well-designed and the proteomics data are presented clearly and concisely. However, the study at this stage seems rather preliminary as it mostly present one descriptive dataset without any follow-ups (see details below).

PLOS authors have the option to publish the peer review history of their article (what does this mean?). If published, this will include your full peer review and any attached files.

Reviewer #1: Yes: Shirit Einav

Reviewer #2: Yes: Pietro Scaturro
---

## [Decision Letter · Decision Letter 1]

20 Dec 2022

Dear Mr. Garishah,

We are pleased to inform you that your manuscript 'Longitudinal proteomic profiling of the inflammatory response in dengue patients' has been provisionally accepted for publication in PLOS Neglected Tropical Diseases.

Best regards,

Simon Rayner

Academic Editor

Sunit Singh

Section Editor

Reviewer's Responses to Questions

**Key Review Criteria Required for Acceptance?**

**Methods**

-Are the objectives of the study clearly articulated with a clear testable hypothesis stated?

-Is the study design appropriate to address the stated objectives?

-Is the population clearly described and appropriate for the hypothesis being tested?

-Is the sample size sufficient to ensure adequate power to address the hypothesis being tested?

-Were correct statistical analysis used to support conclusions?

-Are there concerns about ethical or regulatory requirements being met?

Reviewer #1: Yes

Reviewer #2: (No Response)

**Results**

-Does the analysis presented match the analysis plan?

-Are the results clearly and completely presented?

-Are the figures (Tables, Images) of sufficient quality for clarity?

Reviewer #1: Yes

Reviewer #2: (No Response)

**Conclusions**

-Are the conclusions supported by the data presented?

-Are the limitations of analysis clearly described?

-Do the authors discuss how these data can be helpful to advance our understanding of the topic under study?

-Is public health relevance addressed?

Reviewer #1: Yes

Reviewer #2: (No Response)

**Editorial and Data Presentation Modifications?**

Reviewer #1: (No Response)

Reviewer #2: Accept

**Summary and General Comments**

Reviewer #1: My comments have been addressed.

Reviewer #2: The authors have addressed all my comments, as the revised manuscript provides now a thorough intersection with published studies, a clearer discussion of its intrinsic limitations and an improved description of some of the statistical analysis and methods used.

PLOS authors have the option to publish the peer review history of their article (what does this mean?). If published, this will include your full peer review and any attached files.

Reviewer #1: **Yes: **Shirit Einav

Reviewer #2: **Yes: **Pietro Scaturro

---

## [Editor Report · Acceptance letter]

28 Dec 2022

Dear Dr de Mast,

We are delighted to inform you that your manuscript, "Longitudinal proteomic profiling of the inflammatory response in dengue patients," has been formally accepted for publication in PLOS Neglected Tropical Diseases.

Best regards,

Shaden Kamhawi

co-Editor-in-Chief

Paul Brindley

co-Editor-in-Chief
